# The Role of Insulin Resistance and Diabetes in Nonalcoholic Fatty Liver Disease

**DOI:** 10.3390/ijms21113863

**Published:** 2020-05-29

**Authors:** Hideki Fujii, Norifumi Kawada

**Affiliations:** 1Department of Premier Preventive Medicine, Graduate School of Medicine, Osaka City University, Osaka 545-8585, Japan; rolahideki@med.osaka-cu.ac.jp; 2Department of Hepatology, Graduate School of Medicine, Osaka City University, Osaka 545-8585, Japan

**Keywords:** hepatic fibrosis, insulin resistance, inflammation, stellate cell, hepatocellular carcinoma

## Abstract

Nonalcoholic fatty liver disease (NAFLD) consists of the entire spectrum of fatty liver disease in patients without significant alcohol consumption, ranging from nonalcoholic fatty liver (NAFL) to nonalcoholic steatohepatitis (NASH) to cirrhosis, with NASH recently shown as an important cause of hepatocellular carcinoma (HCC). There is a close relationship between insulin resistance (IR) and NAFLD, with a five-fold higher prevalence of NAFLD in patients with type 2 diabetes (T2DM) compared to that in patients without T2DM. IR is involved in the progression of disease conditions such as steatosis and NASH, as well as hepatic fibrosis progression. The mechanisms underlying these processes involve genetic factors, hepatic fat accumulation, alterations in energy metabolism, and inflammatory signals derived from various cell types including immune cells. In NASH-associated fibrosis, the principal cell type responsible for extracellular matrix production is the hepatic stellate cell (HSC). HSC activation by IR involves “direct” and “indirect” pathways. This review will describe the molecular mechanisms of inflammation and hepatic fibrosis in IR, the relationship between T2DM and hepatic fibrosis, and the relationship between T2DM and HCC in patients with NAFLD.

## 1. Introduction

Nonalcoholic fatty liver disease (NAFLD) is a major form of chronic liver disease that affects both adults and children worldwide [1]. It is one of the clinical consequences of obesity and can progress to nonalcoholic steatohepatitis (NASH), which is pathologically characterized by the presence of steatosis, inflammation, and fibrosis in the liver parenchyma, which ultimately leads to cirrhosis, hepatocellular carcinoma (HCC), and end-stage liver failure [1,2,3]. The prevalence of NAFLD in the world population is estimated to be 25%, whereas the pooled overall NASH prevalence estimated among biopsied NAFLD patients is 59% [4,5]. Recent meta-analysis revealed that over 2146 person-years of follow-up evaluation, 34% had fibrosis progression, 43% had stable fibrosis, and 22% had an improvement in fibrosis stage [6]. The listed diagnosis in the United Network for Organ Sharing database suggests that since 2000, the number of patients with NASH cirrhosis has increased, whereas that of patients with cryptogenic cirrhosis has decreased [7]. Liver-specific mortality and overall mortality among NAFLD and NASH patients were 0.77 and 11.77 per 1000 person-years, and 15.44 and 25.56 per 1000 person-years, respectively [4]. Patients with NAFLD cirrhosis (F4) predominantly develop liver-related events, whereas those with bridging fibrosis (F3) predominantly develop non-hepatic cancers and vascular events [8]. Estimates of NAFLD heritability range from 20% to 70%, with an estimated shared genetic effect or determination between steatosis and fibrosis of 75% [9], depending on ethnicity, study design, environmental factors, and methodology used for NAFLD characterization [10,11].

Recent evidence showed that extra-hepatic organs influence NAFLD progression [12,13,14,15]. Progressive adipose tissue dysfunction and insulin resistance (IR) are key events in NASH development, supporting the existence of an adipose tissue–liver crosstalk [12,13]. Most recently, Vily-Petit et al. reported that increased intestinal gluconeogenesis (IGN) improves glucose control and prevents the onset of hyperglycemia under a high-fat/high-sucrose diet [14]. Conversely, the suppression of IGN promotes lipid accumulation in the liver even under a standard diet. IGN provides metabolic benefits by initiating a gut–brain neural signal, thus triggering brain-dependent regulations of peripheral metabolism [15]. The authors suggested that IGN can specifically modulate the onset of hepatic steatosis and molecular events associated with the development of NAFLD through a gut–brain–liver neural circuit [14].

NAFLD is driven by ectopic fat accumulation in the liver, is an indicator of IR, and signals the possibility of ectopic fat accumulation in inappropriate parts of the body, such as in intramuscular, perivascular, and pericardial regions, and visceral fat deposition [16]. Therefore, NAFLD is often described as the hepatic manifestation of metabolic syndrome and is a major risk factor for type 2 diabetes mellitus (T2DM), in which NAFLD is commonly found as a comorbidity [16,17]. Recent meta-analysis showed that the overall prevalence of NAFLD among patients with T2DM is 55.5%, whereas the global prevalence of NASH among patients with T2DM is 37.3% [18]. Compared to western populations, Asians are particularly susceptible to NAFLD partially because of body composition differences in fat and muscle, as well as genetic susceptibility through the predisposition to T2DM, patatin-like phospholipase domain containing 3 (PNPLA3) SNPs, and polymorphisms in apolipoprotein 3 [3]. T2DM is an important risk factor for NAFLD and seems to accelerate the progression of liver disease in NAFLD patients [1,2,3,4,8,16,17,18]. In this review, we discuss the most updated data characterizing the role of IR and T2DM in the progression of hepatic fibrosis in NAFLD. As described later, improving IR can be one of the potential therapeutic approaches for NASH fibrosis, even in the absence of T2DM. This is an innovative review because we discussed the influence of IR on the progression of hepatic fibrosis in NAFLD, both in basic and clinical aspects, for the first time. Some of the studies reported in this review analyzed data from the Japan Study Group of NAFLD (JSG-NAFLD).

## 2. Molecular Mechanism of Inflammation in IR

Cell death and inflammation are key drivers of fibrosis in NASH and other forms of chronic liver disease [19,20]. The primary pathophysiological mechanisms of IR induced by inflammatory mediators are probably the result of interference with insulin signaling [17]. Insulin acts in all cells by binding to its specific receptor and activating a cascade of intracellular signaling. Upon insulin binding, the insulin receptor phosphorylates itself and several members of the insulin receptor substrate (IRS) family. IRS1 and IRS2 are the main mediators of insulin signaling in the liver, where they control insulin sensitivity [17,21,22]. The canonical IRS signaling pathways include the IRS1- or IRS2-dependent signaling pathways that use the activities of phosphatidylinositol 3-kinase (PI3K)-phosphoinositide-dependent kinase (PDK)-protein kinase B (AKT) and the RAS−extracellular-signal regulated kinase (ERK) [17,21]. The PI3K-PDK-AKT pathway mediates gluconeogenesis and glycogen synthesis. Additionally, the RAS-ERK pathway mediates cell proliferation and survival [21]. IR is defined as the impairment of the appropriate downstream effects of insulin signaling in target tissues, primarily the liver, muscle, and adipose tissue. IR is postulated to begin in muscle tissue, which accounts for up to 70% of glucose disposal, whereas immune-mediated inflammatory change and excess free fatty acids (FFA) cause ectopic lipid deposition [23]. 

IR in adipose tissue results in increased lipolysis in adipocytes and an increased circulating FFA, which further exacerbates steatosis and IR in muscle tissue. During caloric intake, insulin reduces hepatic glucose production by inhibiting glycogenolysis and limiting the postprandial rise in glucose. In IR, this feedback mechanism is impaired and hepatic glucose production continues to rise even when postprandial glucose increases. Glucotoxicity is associated with elevated glucose levels and further contributes to IR [23].

Pro-inflammatory cytokines and transcription factors are highly expressed in various tissues, including adipose tissue or liver in obesity and related disorders. Several inflammatory cytokines, such as tumor necrosis factor (TNF)-α or interleukin (IL)-6, can activate the inhibitor of nuclear factor-κB kinase (IKK) complex IKKβ, c-Jun N-terminal kinase (JNK), which is also known as mitogen-activated protein kinase. Additionally, suppressor of cytokine signaling (SOCS) can phosphorylate IRS1 and IRS2 to inhibit insulin signaling [17,22]. Hyperglycemia can increase oxidative stress [22], which can inhibit insulin signaling through the activation of inhibitor of nuclear factor kappa-B (I-κB) kinase subunit beta (IKKβ) and JNK. IKKβ and JNK can activate nuclear factor kappa-B (NF-κB) and cause its translocation to the nucleus. [17]. Endoplasmic reticulum (ER) stress can also activate the JNK pathway [17,24]. The inflammatory pathways that affect IR are summarized in Figure 1.

From a clinical point of view, data on the changes in hepatic expression of IRS1 and IRS2 with the pathological severity of NAFLD in patients are insufficient and do not provide a consensus. Rametta et al. reported that hepatic *IRS1* mRNA levels did not differ, but *IRS2* mRNA levels progressively increased with the severity of the histology [25]. Conversely, Honma et al. examined mRNA expression in 51 human liver biopsy samples obtained from nondiabetic subjects [26]. The hepatic expression of *IRS1* was unchanged in NAFLD conditions, whereas the expression of *IRS2* in NASH samples was lower than that in health control samples. Moreover, the severity of lobular inflammation was negatively correlated with hepatic *IRS2* mRNA levels (r = −0.35, *p* < 0.05); by contrast, *IRS1* mRNA levels were not significantly correlated with the severity of any of the histological features examined [26]. Enooku et al. examined 146 biopsy-proven NAFLD samples and revealed that *IRS1* mRNA levels decreased with increasing degrees of hepatic necroinflammatory activity; however, *IRS2* mRNA levels were not significantly correlated with this activity [27]. To date, the effects of changes in hepatic *IRS1*/*IRS2* mRNA levels on hepatic inflammation in NAFLD patients have not been resolved. 

## 3. Genetic Factors Affecting IR and NAFLD 

To date, at least five variants in different genes have been found to be strongly associated with the susceptibility to and progression of NAFLD, namely, PNPLA3, transmembrane 6 superfamily member 2 (TM6SF2), glucokinase regulator (GCKR), membrane bound O-acyltransferase domain-containing 7 (MBOAT7), and hydroxysteroid 17β-dehydrogenase (HSD17B13) [10]. First, the rs738409 C>G SNP, which results in the I148M protein variant of PNPLA3, has been linked to higher hepatic fat content but with no major effects on IR and adiposity features [10,28]. Second, the rs58542926 C>T SNP, which encodes for the E167K variant TM6SF2, shows higher hepatic and adipose tissue IR and enhanced muscle insulin sensitivity compared to CC homozygotes [29]. Third, the NAFLD risk variant GCKR (P446L) is associated with higher levels of plasma low-density lipoprotein cholesterol and triglycerides and lower fasting glucose and homeostasis model assessment parameter (HOMA)-IR [30]. Fourth, MBOAT7 suppression by obesity or the MBOAT7 rs641738 variant promotes NAFLD and IR progression [31]. Downregulation of MBOAT7 by hyperinsulinemia contributes to fatty liver independently of genetic background [32]. The rs626283 polymorphism in MBOAT7 is associated with NAFLD and impaired insulin sensitivity in obese children and adolescents [33]. Finally, the rs72613567 variant in HSD17B13 plays a role in regulating retinoic acid metabolism, suggesting that retinol may be involved in NAFLD development [10].

## 4. Molecular Mechanism of Hepatic IR Affects Hepatic Fibrosis

Fibrosis is the result of excessive production of extracellular matrix (ECM) that is not adequately maintained and, thus, results in net accumulation. In the liver, hepatic stellate cells (HSCs) constitute the main source of ECM-producing fibroblasts in models of toxic and biliary liver disease and NAFLD [19,34]. IR is recognized as an integral component of NAFLD pathogenesis that worsens with disease progression [19,20,21]. Our review suggests that the activation of HSC by IR is largely divided into distinct direct and indirect pathways. 

### 4.1. Indirect Pathway 

Inflammation can be induced by IR itself. Hepatocyte stress and death also promote inflammation, leading to the recruitment of macrophages and secretion of profibrogenic mediators such as transforming growth factor-β (TGF-β), which is the center of the fibrogenic response in NASH [19,20,34,35]. There is also strong evidence for HSC activation through the direct interactions of stressed or dead hepatocytes with HSCs. This may be through the release of profibrogenic damage-associated molecular patterns [36,37] or other profibrogenic mediators such as Hh ligands and osteopontin (OPN) [19], or via apoptotic bodies [38,39] that may directly act on HSCs. High mobility group box 1 released by necrotic hepatocytes also mediates the recruitment of neutrophils in an acetaminophen-induced liver injury model through an interaction with a receptor for advanced glycation end products [40]. C–C chemokine receptors type 2 (CCR2) and 5 (CCR5) and their ligands CCL2 and CCL5 are implicated in the pathogenesis of liver inflammation and fibrosis, especially in NASH [35]. In response to hepatocyte injury, Kupffer cells secrete CCL2, which recruits monocytes to the liver. Inhibition of CCL2/CCR2 or CCL5/CCR5 has been shown to attenuate liver fibrosis in mice [41,42,43], and a recent phase 2b clinical trial showed that treatment with the CCR2/5 antagonist cenicriviroc resulted in an early antifibrotic benefit that was maintained particularly in the subset of NASH patients with advanced fibrosis [44]. Transcriptional coactivator with PDZ binding motif (TAZ) is a paralogue of YAP and key component of the HIPPO-YAP/TAZ-TAZ/TEA domain (TEAD) signaling cascade that is strongly upregulated in hepatocytes in both mouse models and NASH patients [45]. However, TAZ was not upregulated in simple steatosis, suggesting that TAZ could be involved in the transition from simple steatosis to NASH. Hepatocyte TAZ to NASH fibrosis is a TEAD-mediated induction of the secretory factor Indian hedgehog that activates fibrogenic genes in HSCs [45]. The Notch signaling pathway is important for multiple cell differentiation processes during embryonic and adult stages [19]. Notch activity is substantially increased in murine and human NASH [46]. In addition, hepatocyte Notch activation induces Sox9-dependent OPN secretion, which can directly activate HSCs, independent of hepatocellular injury, leading to collagen deposition [46]. Notch activation also increases FoxO1 activation at gluconeogenic promoters, leading to glucose intolerance [47]. The pharmacological blockade of Notch signaling by γ-secretase inhibitors improves glucose tolerance and IR [48]. Dongiovanni et al. reported that IR promotes ECM stabilization by a mechanism encompassing overexpression of lysyl oxidase-like 2 (LOXL2) in response to lipotoxicity [49]. Importantly, hepatic LOXL2 upregulation was specifically detected in NAFLD patients with T2DM progressing to advanced fibrosis [49]. LOXL2 may be a new therapeutic target in chronic liver diseases [50]. Adiponectin, an adipokine that modulates several metabolic processes including glucose metabolism and fatty acid oxidation, acts as an anti-inflammatory cytokine and inhibits the development of IR and NAFLD [51,52]. Adiponectin inhibits the differentiation of myeloid progenitor cells and modulates Kupffer cell function via reduction of Toll-like receptor 4 signaling [52]. Adiponectin inhibits the production of inflammatory cytokines such as TNF-α, monocyte chemotactic protein-1, and IL-6, and suppresses proinflammatory classically activated (M1) macrophage activation [51,52]. In addition, adiponectin upregulates the production of the anti-inflammatory cytokine IL-10 in macrophage and promotes anti-inflammatory alternatively activated (M2) macrophage proliferation [51,52]. The indirect pathways regulating HSC activation in steatohepatitis are summarized in Figure 2.

### 4.2. Direct Pathway 

A small number of reports suggest that hyperinsulinemia and hyperglycemia directly activate HSCs. A study showed that IR-related hyperinsulinemia can directly stimulate HSCs to proliferate and secrete type I collagen by differentially activating PI3K and ERK-dependent pathways [53]. Ota et al. reported that a high-fat diet induced IR and increased the expression of the profibrotic TGF-β1 in obese rats [54]. Recently, another report suggested that insulin-like growth factor 1 (IGF1) binds its receptor (IGF1R) in HSCs to promote IGF1R via IRS2 to trigger ERK1/2 phosphorylation, which leads to the expression of matrix metalloproteinase (MMP) 9 [55].

High serum glucose during hyperglycemia can lead to the activation of the HSCs [56]. Kiss et al. reported that high glucose exposure to LX-2 cells (e.g., an immortalized human HSC cell lineage) resulted in decreased MMP2 activity and deceleration of type I collagen in the ER, with decreased pS6 expression pointing to development of ER stress [57]. Hyperglycemia can also aggravate hepatic fibrosis, which may be associated with HSC autophagy induced by acid-sensing ion channel 1a, which is a subfamily of the degenerin/epithelial Na^+^ channel family of the non-voltage-gated cation channel that acts through the CaMKKβ/ERK signal pathway [58]. Although it remains unclear whether HSCs are directly activated in prediabetic status, further study will be needed because many NAFLD patients are prediabetes. The direct pathways regulating HSC activation in steatohepatitis are summarized in Figure 3.

## 5. Relationship Between T2DM and Hepatic Fibrosis in Patients with NAFLD 

### 5.1. Liver Biopsy

The gold standard for diagnosing NASH remains a liver biopsy [1]. Many past studies have confirmed that the presence of T2DM is an independent predictor of advanced hepatic fibrosis (F3 or F4) in biopsy-diagnosed patients with NAFLD [1,2,3,4,8,16,17,18]. For example, we previously reported that T2DM was an independent predictor of advanced fibrosis in patients with NAFLD (Odds ratio (OR) 2.9, 95% confidence interval (CI) 1.3–6.1, *p* = 0.007) [59]. An earlier study from JSG-NAFLD showed that in 1365 biopsy-proven Japanese NAFLD patients, 47.3% had T2DM, whereas the multivariate analysis revealed that the presence of T2DM was one of the risk factors for advanced fibrosis (OR 2.39, 95% CI 1.60–3.55, *p* < 0.001) [60]. A report from Hong Kong showed that in 94 patients with T2DM who underwent liver biopsy, 56% had steatohepatitis and 50% had advanced hepatic fibrosis [61]. A more recent, prospective multicenter study showed that in a cohort of 458 NAFLD patients with advanced fibrosis, 67% had T2DM, suggesting that T2DM is a robust predictor of poor transplantation-free survival (hazard ratio (HR) 3.33) and liver-related outcomes [8]. With the increase in the number of clinical trials, the number of observational studies of biopsy-proven NAFLD has decreased.

### 5.2. Serum Biomarkers

Several noninvasive biomarkers have been proposed for NASH detection and to avoid redundant liver biopsies. In western countries, serum biomarkers for staging fibrosis include predictive models (e.g., NAFLD fibrosis score (NFS) [62]) and direct measures of fibrosis (e.g., PIIINP [63] or Pro-C3 [64]) to discriminate between patients with advanced fibrosis [65]. Some of these markers were originally designed for hepatitis C using the aspartate transaminase (AST)/alanine transaminase ratio, the aspartate transaminase-to-platelet ratio index [66], and Fibrosis-4 (FIB-4) [67]. Yoneda et al. from JSG-NAFLD reported that the platelet (PLT) count is the simplest index for predicting advanced fibrosis in patients with NAFLD [68]. Indeed, PLT levels may be unexpectedly high even when hepatic fibrosis is advanced; for example, patients with stage 3 hepatic fibrosis had a PLT of 189 × 10^9^/L, whereas patients with stage 4 had 153 × 10^9^/L [68]. 

In Japan, fibrosis markers have also been extensively examined, including hyaluronic acid, type IV collagen 7S, procollagen III peptide, and *Wisteria floribunda* agglutinin-positive Mac-2-binding protein (WFA^+^-M2BP). WFA^+^-M2BP is a novel serum fibrosis biomarker for chronic hepatitis C that is clinically validated and covered by the Japanese health insurance system [69]. WFA^+^-M2BP is also used for assessing liver fibrosis in patients with NAFLD [70,71]. Kamada et al. reported that serum levels of Mac-2 binding protein (Mac-2bp) can be used to predict the histologic severity of hepatic fibrosis in patients with NAFLD [72,73]. Furthermore, they tried a head-to-head comparison between WFA^+^-M2BP and Mac-2bp in which 510 patients with NAFLD from JSG-NAFLD were used. WFA^+^-M2BP and Mac-2bp were equally useful for NASH diagnosis, although Mac-2bp was superior to WFA^+^-M2BP for the prediction of the NAFLD fibrosis stage, especially for early stages fibrosis (F1 and F2) [74]. The measurement of autotaxin (ATX) has been covered by the national health insurance in Japan since June 2018 as an auxiliary method of quantifying the degree of liver fibrosis in patients with chronic liver disease and cirrhosis [75]. Some groups found that the serum ATX concentration significantly correlates with the fibrosis stage in NAFLD patients [75,76,77]. Recently, Okanoue et al. suggested that the combination of type IV collagen 7S and AST (CA index-NASH = 0.994×type IV collagen 7S + 0.0255×AST) is a reliable and simple scoring system to diagnose advanced fibrosis. The area under the receiver operating characteristics (AUROCs) for training/validation data sets are 0.842/0.931 for NASH-related advanced fibrosis [78]. 

The NFS and FIB-4 have been externally validated in populations of different ethnicities with consistent results. Singh et al. investigated a total of 1319 patients with T2DM who underwent liver biopsy for suspected NAFLD [79]. The diagnostic abilities were FIB-4 > 2.67 and NFS > 0.676 advanced fibrosis, indicating reasonable specificities of 69.9% and 93% but poor sensitivities of 6.7% and 44.1%, respectively. The AUROCs used to detect advanced fibrosis were 0.77 and 0.72, respectively. Another report from Turkey examined 349 patients with biopsy-proven NAFLD (166 with T2DM) and showed that a FIB-4 with a low cutoff value of 1.3 had a specificity of 67% in patients with T2DM and 69% in those without [80]. Conversely, a FIB-4 with a high cutoff value of 2.67 had a sensitivity of 22% in patients with T2DM and 0% in those without [80]. NFS performed similar to FIB-4, suggesting that both FIB-4 and NFS have limited utility in diagnosing advanced fibrosis in NAFLD, especially in patients with T2DM.

### 5.3. Vibration-Controlled Transient Elastography 

There have been some attempts to estimate the prevalence of NASH by noninvasive methods [1]. Vibration-controlled transient elastography (VCTE) is a rapid, safe, and reproducible procedure for liver stiffness measurement (LSM) that can be performed at the bedside with immediate results [81]. VCTE generally provides an accurate per-protocol risk assessment of advanced fibrosis (F3 or F4) in NAFLD (AUROCs ranging from 0.80 to 0.94) [65,82,83]. Several meta-analyses, mostly performed in viral hepatitis patients, reported good (88–89%) and excellent (93–96%) TE accuracy for diagnosing advanced fibrosis and cirrhosis, respectively [65,84]. A recent large prospective diagnostic study from the United Kingdom demonstrated that LSM identified patients with fibrosis from AUROCs of 0.77 (95% CI 0.72–0.82) for ≥ F2, 0.80 (95% CI 0.75–0.84) for ≥ F3, and 0.89 (95% CI 0.84–0.93) for F4. The Youden cutoff values for F2, F3, and F4 were 8.2, 9.7, and 13.6 kPa, respectively [85]. Probe type (M or XL probe) did not affect LSM; Oeda et al. also confirmed that liver fibrosis and steatosis could be equally evaluated with M and XL probes in patients with NAFLD [86]. Roulot et al. examined 705 French diabetic patients and reported that 12.7% (N = 85) had LSM ≥ 8 kPa, which was suggestive of significant fibrosis, whereas 2.1% (N = 14) had LSM ≥ 13 kPa, which indicated cirrhosis [87]. In the Rotterdam study, a population-based study among individuals ≥ 45 y combined the presence of T2DM and hepatic steatosis (OR, 5.20; 95% CI: 3.01–8.98; *p* < 0.001 for combined presence), which were associated with LSM ≥ 8.0 kPa in multivariable analyses [88]. These studies clearly demonstrated that the presence of T2DM, especially that concurrent with steatosis, resulted in increased probabilities of having clinically relevant fibrosis. 

### 5.4. Magnetic Resonance Elastography

Magnetic resonance elastography (MRE) is not operator-dependent or affected by obesity or ascites. It is currently the most accurate imaging tool for the detection of liver fibrosis because it is more effective than TE alone or TE with serum biomarkers combined [65,89,90]. For example, Imajo et al. examined 142 patients with biopsy-proven NAFLD in Japan who had an average body mass index (BMI) of 28.1 kg/m^2^ and underwent VCTE using M-probe, as well as MRE. The authors found that MRE was better than VCTE for detecting fibrosis [89]. Doycheva et al. performed a cross-sectional analysis of 100 patients with T2DM and reported that the prevalence of NAFLD defined as MRI- proton-density fat traction ≥ 5% and advanced fibrosis defined as MRE ≥ 3.6 kPa was 65% and 7.1%, respectively [91]. A recent study using MRE to evaluate fibrosis revealed that the prevalence of F2 (≥ 3.0 kPa) and F3 (≥ 3.6 kPa) in the overall cohort was 5.1% and 1.3%, respectively [92]. Additionally, the prevalence of F2 and F3 in participants with NAFLD plus T2DM was 24.1% and 6.0%, respectively. In a multivariate analysis of this cohort, only age, insulin, T2DM, and fatty liver on MR were independently associated with significant fibrosis [92]. Limitations of MRE include cost and availability, along with patient-dependent factors such as the presence of magnetically susceptible implants, being able to hold their breath, and claustrophobia [81]. Iron overload, high BMI, and significant ascites were also associated with technical failure [81,93]. The benefit of MRE is that it allows for a much larger sampling compared to US techniques and liver biopsy, which may reduce sampling variability secondary to heterogeneity of fibrosis [94]. The utility of MRE in NAFLD patients is promising, but further validation is required [65,95]. 

## 6. The Role of IR in Hepatic Fibrosis in NAFLD Patients 

In 2014, Jung et al. reported that the value of the HOMA-IR was significantly higher in NASH subjects than in healthy controls (4.4 ± 2.5 vs. 1.7 ± 0.6; *p* < 0.001), but that this was not the case in patients with T2DM [96]. Unfortunately, their study could not show a significant correlation of HOMA-IR to the severity of either histologic grading or staging, because of the small sample size (41 cases of biopsy-proven NAFLD). Kessoku et al. evaluated 1365 biopsy-proven NAFLD patients registered in the JSG-NAFLD database and showed that HOMA-IR significantly increased depending on the degree of hepatic fibrosis (2.7 ± 1.1 in stage 0, 3.5 ± 1.6 in stage 1, 4.0 ± 1.7 in stage 2, 4.3 ± 1.7 in stage 3, and 4.6 ± 2.1 in stage 4; *p* < 0.01) [97]. Ballestri et al. in Italy investigated 118 consecutive biopsy-proven NAFLD patients (25% with T2DM) and reported that HOMA-IR independently predicted advanced hepatic fibrosis [98]. In both studies, the subjects included patients with T2DM. In fact, the prevalence of T2DM among patients with NAFLD has been reported to be 22.51% in global studies [4]. Conversely, approximately 80% of NAFLD patients are nondiabetic; therefore, can HOMA-IR also act as an independent predictor of advanced fibrosis even in nondiabetic patients? We recently investigated 361 biopsy-proven Japanese NAFLD patients without T2DM who were registered with JSG-NAFLD. We reported that HOMA-IR ≥ 2.90 was an independent predictor of advanced fibrosis in nondiabetic NAFLD patients, and our data suggest that there may be a pathway for IR to directly activate HSCs [99]. Unfortunately, there are no US Food and Drug Administration-approved medications to treat NAFLD and guidelines for NAFLD management are not well established. In this context, improving IR may be beneficial for anti-fibrotic treatment, even among patients with nondiabetic NAFLD [100,101].

## 7. Hepatocellular Carcinoma

The exact pathogenesis of HCC in NAFLD has not been fully described; however, both obesity and T2DM seem to play a critical role in hepatocarcinogenesis [102,103]. The global annual incidence of HCC in people with NAFLD is estimated to be 0.44 per 1000 person-years, whereas for those with NASH, the incidence of HCC is higher at 5.29 per 1000 person-years, which most likely results from the inflammatory state in those with NASH that promotes fibrosis and disease progression [4]. Tokushige et al. from JSG-NAFLD examined 532 patients with alcoholic liver disease (ALD)-HCC and 209 patients with NAFLD-HCC and revealed that the prevalence of lifestyle-related diseases including T2DM and hypertension was higher in the NAFLD-HCC group than in the ALD-HCC group [104]. In one study using data from the Surveillance, Epidemiology, and End Results database that covered a six-year period of time from 2004 to 2009, investigators determined that there was a 9% annual increase in the number of HCC cases attributed to NAFLD [105]. In another study using data from four European primary care databases representing the UK, Netherlands, Italy, and Spain, the authors reported that coded NAFLD/NASH patients were more likely to have T2DM, hypertension, and obesity than the matched controls [106]. The HR for HCC in patients compared to those in controls was 3.51 (95% CI 1.72–7.16). The strongest independent predictor of an HCC or cirrhosis diagnosis was baseline diagnosis of T2DM [106]. A recent study demonstrated with a multivariate analysis of 354 patients with NASH cirrhosis that T2DM was associated with an increased risk of developing HCC (HR 4.2; 95% CI 1.2–14.2) [107]. Although the risk of HCC is higher in NASH-related cirrhosis, there is also evidence that HCC can occur in the absence of cirrhosis (26–37%) in both NAFLD and NASH [108,109,110]. Bengtsson et al. examined the mortality rate and concluded that the parameters that independently associate with increased mortality included the Barcelona Clinic Liver Cancer stage, number of tumors, lower albumin, and presence of T2DM [110]. 

A number of genetic factors have also been implicated in the development of NAFLD. PNPLA3 polymorphisms were noted to not only lead to an increased risk of steatohepatitis and fibrosis, but also of a three-fold increased risk of HCC. These risks were independent of age, gender, BMI, T2DM, and presence of fibrosis or cirrhosis [103]. However, whether carrying the transmembrane 6 superfamily member 2 (TM6SF2; rs58542926 c.449 C>T, p.E167K) polymorphism is an independent HCC risk factor is still controversial [103,111,112]. 

## 8. Conclusions

In this article, several aspects that potentially contribute to the mechanisms underlying IR and T2DM in the progression of NAFLD are discussed. We also discussed the indirect/direct roles of IR during the progression of hepatic fibrosis. More detailed knowledge of these mechanisms in NAFLD are needed to identify novel therapeutic approaches for this disease. Furthermore, specific biomarkers that can predict the degree of liver damage and NAFLD progression to cirrhosis and cancer should be developed. To achieve these objectives, we need animal models that accurately reflect the metabolic and histological characteristics of human NAFLD. Further investigations using such models can provide additional insight into the underlying mechanism by which extrahepatic organs (e.g., gut, adipose tissue, brain) regulate IR. 

## Figures and Tables

**Figure 1 ijms-21-03863-f001:**
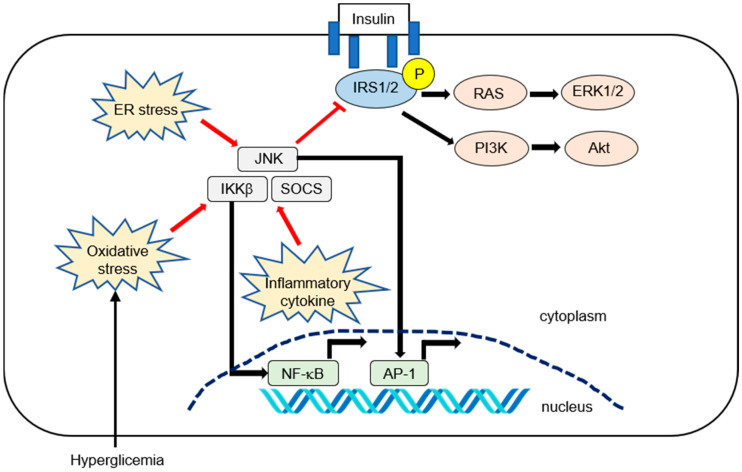
Summary of inflammatory pathways affecting hepatic insulin resistance (IR) in nonalcoholic fatty liver disease (NAFLD). Insulin activates its receptor, which results in tyrosine phosphorylation on the insulin receptor substrate (IRS1 and IRS2) and activation of downstream effector pathways, including the phosphatidylinositol 3-kinase (PI3K)-phosphoinositide-dependent kinase (PDK)-protein kinase B (AKT) and the RAS−extracellular-signal-regulated kinase (ERK) pathways (i.e., canonical IRS signaling). Numerous pro-inflammatory signaling or reactive oxygen species can activate IKK-β. The activated NF-kB is then translocated into the nucleus and binds to specific DNA response elements. Inflammatory cytokines such as IL-6 promote IR by inducing suppressor of cytokine signaling (SOCS) 1 and 3. SOCS1 and SOCS3 impair insulin signaling through ubiquitin-dependent degradation of IRS. The c-Jun N-terminal kinase (JNK, or mitogen-activated protein kinase) represents another important inhibitory kinase of IRS that is activated in response to a variety of extracellular stimuli and cellular stressors such as oxidative and endoplasmic reticulum (ER) stress.

**Figure 2 ijms-21-03863-f002:**
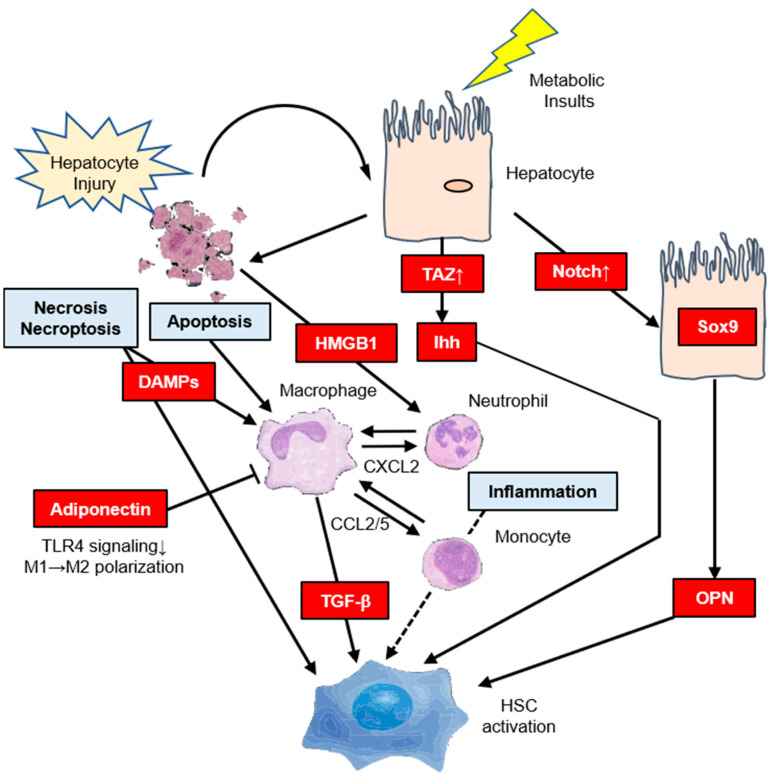
The indirect pathways regulating hepatic stellate cell (HSC) activation in steatohepatitis. Metabolic insults such as hyperinsulinemia/hyperglycemia lead to the activation of transcriptional coactivator with PDZ-binding motif (TAZ) in hepatocytes. Increased hepatocyte expression of TAZ in nonalcoholic steatohepatitis (NASH) but not simple steatosis directly leads to HSC activation through the release of Indian hedgehog (Ihh) and promotes hepatocyte injury and inflammation that may indirectly promote HSC activation. Notch activation by cell-surface ligands on a neighboring cell leads to a Sox9-dependent increase in osteopontin (OPN) secretion to activate HSCs. HSC activation occurs through direct interactions between stressed or dead hepatocytes (i.e., apoptosis, necrosis, or necroptosis) and HSCs. This may be through the release of profibrogenic damage-associated molecular patterns. High mobility group box 1 (HMGB1) is released by injured hepatocytes to mediate the recruitment of neutrophils. Monocyte infiltration into the liver is primarily controlled by C–C chemokine receptors (CCR2) and its ligand CCL2, which may serve as therapeutic targets in NASH. Adiponectin modulates Kupffer cell function via reduction of Toll-like receptor 4 (TLR4) signaling, and directly stimulates M1→M2 polarization.

**Figure 3 ijms-21-03863-f003:**
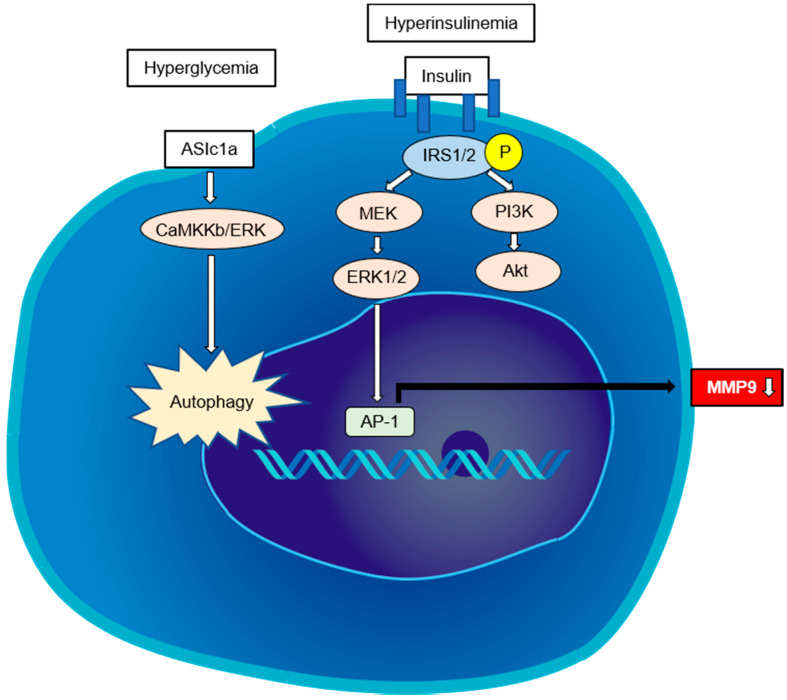
The direct pathways regulating HSC activation in steatohepatitis. Hyperinsulinemia can directly stimulate HSCs to proliferate and secrete type I collagen by differentially activating PI3K- and ERK-dependent pathways. IGF1R triggers ERK1/2 phosphorylation via IRS2, leading to the expression of matrix metalloproteinase (MMP)-9. Hyperglycemia can aggravate hepatic fibrosis, which may be associated with the HSC autophagy induced by acid-sensing ion channel 1a (ASIC1a).

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
