# Peer review of "The Role of Insulin Resistance and Diabetes in Nonalcoholic Fatty Liver Disease"

_ijms, 2020, doi:10.3390/ijms21113863_

Round 1

Reviewer 1 Report

Fujii and colleagues describe the role of insulin resistance and diabetes in NAFLD onset and progression. The topic is certantly of great interest, however several reviews of the literature previously addressed it.Thus, to ameliorate the draft of the manuscript and making it more appealing, i suggest to introduce novel findinds about recent studies and 'open debates'. In particular, an extent revision of the abstract and concluding remarks may grab the reader's attention. Moreover, in the introduction some references about the primary inherited factors could be of interest. Specifically, several reviews and recent papers studies the effect of genetic variations on the entire spectrum of NAFLD and the link to IR (i.e. PNPLA3, TM6SF2, MBOAT7, GCKR - Qiao A, (PMID: 21547936), Musso G (PMID: 28242789), Speliotes EK (PMID: 21423719), Santoro N (PMID: 26043229)Meroni (PMID:32058943); Helsley (PMID:31621579), Umano (PMID:29485130). A short paragraph about the effect of hyperinsulinemia and IR on genetic factors should be of interest. There are very novel papers about the interplay between IR, fatty liver and genetics (See above).

The review, moreover, should benefit also of a particular hint about the effect of Notch signaling ectopic activation during steatohepatitis and the link with progressive liver damage (i.e. Schwabe RF (PMID: 32044315);Zhou Y (PMID: 32275903), Zhu C (PMID:30463916) Morell CM (PMID:26641143)).

Finally, a comment about different point of views on NAFLD could be important i.e. Vily-Petit J (PMID: 32205419), Mann (PMID: 26638195), Dongiovanni (PMID: 28468951) Overi (PMID: 32131439).

Author Response

We thank the editor and reviewers for the positive assessment of our manuscript and for identifying areas that required correction or modification. Please note that the changes made do not influence the content, conclusions, or framework of the paper. We have not listed below all minor changes made; however, these are indicated in red font in the revised manuscript. All line numbers mentioned in the response to each comment refer to the numbers that appear at the left margin of the text in the revised manuscript.

Reviewer’s Comments to Author:
Reviewer 1

  1. Fujii and colleagues describe the role of insulin resistance and diabetes in NAFLD onset and progression. The topic is certantly of great interest, however several reviews of the literature previously addressed it. Thus, to ameliorate the draft of the manuscript and making it more appealing, I suggest to introduce novel findings about recent studies and 'open debates'. In particular, an extent revision of the abstract and concluding remarks may grab the reader's attention. Moreover, in the introduction some references about the primary inherited factors could be of interest. Specifically, several reviews and recent papers studies the effect of genetic variations on the entire spectrum of NAFLD and the link to IR (i.e. PNPLA3, TM6SF2, MBOAT7, GCKR - Qiao A, (PMID: 21547936), Musso G (PMID: 28242789), Speliotes EK (PMID: 21423719), Santoro N (PMID: 26043229), Meroni (PMID:32058943); Helsley (PMID:31621579), Umano (PMID:29485130). A short paragraph about the effect of hyperinsulinemia and IR on genetic factors should be of interest. There are very novel papers about the interplay between IR, fatty liver and genetics (See above).

Response: Thank you for the valuable comments. We made four modifications in accordance with the reviewer’s comments.

1) We have added the word “genetic factors” in the Abstract (page 1, line 19).

2) We have added the following text to the Introduction (page 2, line 46–49):

Estimates of NAFLD heritability range from 20% to 70%, with an estimated shared genetic effect or determination between steatosis and fibrosis of 75% [1], depending on the ethnicity, study design, environmental factors, and methodology used for NAFLD characterization [2,3].

References

  1. Cui, J., Chen, C. H., Lo, M. T., Schork, N., Bettencourt, R., Gonzalez, M. P., Bhatt, A., Hooker, J., Shaffer, K., Nelson, K. E., et al. For The Genetics Of Nafld In Twins, C., Shared genetic effects between hepatic steatosis and fibrosis: A prospective twin study. Hepatology 2016, 64:1547-1558.
  2. Eslam, M., George, J. Genetic contributions to NAFLD: leveraging shared genetics to uncover systems biology. Nat Rev Gastroenterol Hepatol 2020, 17:40-52.
  3. Eslam, M., Valenti, L., Romeo, S. Genetics and epigenetics of NAFLD and NASH: Clinical impact. J Hepatol 2018, 68:268-279.

3) We have added a short paragraph to Section 3. Genetic factors affecting IR and NAFLD (page 5, line 139–155):

To date, at least five variants in different genes have been strongly associated with

the susceptibility to and progression of NAFLD, namely, PNPLA3,

transmembrane 6 superfamily member 2 (TM6SF2), glucokinase regulator (GCKR),

membrane bound O- acyltransferase domain- containing 7 (MBOAT7) and

hydroxysteroid 17β- dehydrogenase (HSD17B13) [2]. The rs738409 C>G SNP, which results in the I148M protein variant of PNPLA3, has been linked to higher hepatic fat content but without a major direct effect on IR and adiposity features [2,4]. The rs58542926 C>T SNP, which encodes for the E167K variant TM6SF2, shows higher hepatic and adipose tissue IR and enhanced muscle insulin sensitivity as compared to CC homozygotes [5]. The NAFLD risk variant GCKR (P446L) is associated with higher levels of plasma low-density lipoprotein cholesterol and triglycerides and lower fasting glucose and homeostasis model assessment parameter (HOMA)-IR [6]. MBOAT7 suppression by obesity or due to the MBOAT7 rs641738 variant promotes NAFLD and IR progression [7]. Moreover, down-regulation of MBOAT7 by hyperinsulinemia contributes to fatty liver independently of the genetic background [8]. The rs626283 polymorphism in MBOAT7 is associated with NAFLD and impaired insulin sensitivity in obese children and adolescents [9]. In addition, the rs72613567 variant in HSD17B13 plays a role in the regulation of retinoic acid metabolism, suggesting that retinol may be involved in NAFLD development [2].

References

  1. Eslam, M., George, J. Genetic contributions to NAFLD: leveraging shared genetics to uncover systems biology. Nat Rev Gastroenterol Hepatol 2020, 17:40-52.
  2. Romeo, S., Kozlitina, J., Xing, C., Pertsemlidis, A., Cox, D., Pennacchio, L. A., Boerwinkle, E., Cohen, J. C., Hobbs, H. H. Genetic variation in PNPLA3 confers susceptibility to nonalcoholic fatty liver disease. Nat Genet 2008, 40:1461-5.
  3. Musso, G., Cipolla, U., Cassader, M., Pinach, S., Saba, F., De Michieli, F., Paschetta, E., Bongiovanni, D., Framarin, L., Leone, N., et al. TM6SF2 rs58542926 variant affects postprandial lipoprotein metabolism and glucose homeostasis in NAFLD. J Lipid Res 2017, 58:1221-1229.
  4. Speliotes, E. K., Yerges-Armstrong, L. M., Wu, J., Hernaez, R., Kim, L. J., Palmer, C. D., Gudnason, V., Eiriksdottir, G., Garcia, M. E., Launer, L. J., et al. Genome-wide association analysis identifies variants associated with nonalcoholic fatty liver disease that have distinct effects on metabolic traits. PLoS Genet 2011, 7:e1001324.
  5. Helsley, R. N., Varadharajan, V., Brown, A. L., Gromovsky, A. D., Schugar, R. C., Ramachandiran, I., Fung, K., Kabbany, M. N., Banerjee, R., Neumann, C. K., et al. Obesity-linked suppression of membrane-bound O-acyltransferase 7 (MBOAT7) drives non-alcoholic fatty liver disease. Elife 2019, 8. pii: e49882. doi: 10.7554/eLife.49882. 8.
  6. Meroni, M., Dongiovanni, P., Longo, M., Carli, F., Baselli, G., Rametta, R., Pelusi, S., Badiali, S., Maggioni, M., Gaggini, M., et al. Mboat7 down-regulation by hyper-insulinemia induces fat accumulation in hepatocytes. EBioMedicine 2020, 52:102658.
  7. Umano, G. R., Caprio, S., Di Sessa, A., Chalasani, N., Dykas, D. J., Pierpont, B., Bale, A. E., Santoro, N. The rs626283 Variant in the MBOAT7 Gene is Associated with Insulin Resistance and Fatty Liver in Caucasian Obese Youth. Am J Gastroenterol 2018, 113:376-383.

4) We have added the following sentence to the Conclusion (page 13, line 406–408):

Further investigations with such models are predicted to provide additional insight into the mechanism by which extrahepatic organs (e.g., gut, adipose tissue, brain) regulate IR.

 2. The review, moreover, should benefit also of a particular hint about the effect of Notch signaling ectopic activation during steatohepatitis and the link with progressive liver damage (i.e. Schwabe RF (PMID: 32044315);Zhou Y (PMID: 32275903), Zhu C (PMID:30463916) Morell CM (PMID:26641143)). Response: Thank you for the valuable comments. As the reviewer has pointed out, we discussed Notch signaling and LOXL2 as follows (page 7, line 203–214):The Notch signaling pathway is important for multiple cell differentiation processes during embryonic and adult life [10]. Notch activity is substantially increased in murine and human NASH [11]. In addition, hepatocyte Notch activation induces Sox9-dependent OPN secretion, which can directly activate HSCs independent of hepatocellular injury, leading to collagen deposition [11]. Notch activation also increases FoxO1 activation at gluconeogenic promoters,leading to glucose intolerance [12]. Pharmacological blockade of Notch signaling by γ -secretase inhibitors improved glucose tolerance and IR [13]. Dongiovanni et al. reported that IR promotes ECM stabilization through a mechanism encompassing overexpression of lysyl oxidase-like 2 (LOXL2) in response to lipotoxicity [14]. Importantly, hepatic LOXL2 up-regulation was specifically detected in NAFLD patients with T2DM progressing to advanced fibrosis [14]. Thus, LOXL2 may be a new therapeutic target in chronic liver diseases [15].

References

  1. Schwabe, R. F., Tabas, I., Pajvani, U. B. Mechanisms of Fibrosis Development in NASH. Gastroenterology 2020.
  2. Zhu, C., Kim, K., Wang, X., Bartolome, A., Salomao, M., Dongiovanni, P., Meroni, M., Graham, M. J., Yates, K. P., Diehl, A. M. Hepatocyte Notch activation induces liver fibrosis in nonalcoholic steatohepatitis. Sci Transl Med 2018, 10:468.
  3. Pajvani, U. B., Shawber, C. J., Samuel, V. T., Birkenfeld, A. L., Shulman, G. I., Kitajewski, J., Accili, D. Inhibition of Notch signaling ameliorates insulin resistance in a FoxO1-dependent manner. Nat Med 2011, 17:961-7.
  4. Dongiovanni, P., Rametta, R., Meroni, M., Valenti, L. The role of insulin resistance in nonalcoholic steatohepatitis and liver disease development--a potential therapeutic target? Expert Rev Gastroenterol Hepatol 2016, 10:229-42.
  5. Dongiovanni, P., Meroni, M., Baselli, G. A., Bassani, G. A., Rametta, R., Pietrelli, A., Maggioni, M., Facciotti, F., Trunzo, V., Badiali, S., et al. Insulin resistance promotes Lysyl Oxidase Like 2 induction and fibrosis accumulation in non-alcoholic fatty liver disease. Clin Sci (Lond) 2017, 131:1301-1315.
  6. Ranjbar, G., Mikhailidis, D. P., Sahebkar, A. Effects of newer antidiabetic drugs on nonalcoholic fatty liver and steatohepatitis: Think out of the box! Metabolism 2019, 101:154001.

  1. Finally, a comment about different point of views on NAFLD could be important i.e. Vily-Petit J (PMID: 32205419), Mann (PMID: 26638195), Dongiovanni (PMID: 28468951) Overi (PMID: 32131439)

Response: We have added the following text to the manuscript (page 2, line 50-60):

Recent evidence revealed that extra-hepatic organs influence the progression of NAFLD. Progressive adipose tissue dysfunction and insulin resistance (IR) represent key events in NASH development, supporting the existence of adipose tissue–liver crosstalk [16,17]. Most recently, Vily-Petit et al. reported that increased intestinal gluconeogenesis (IGN) improves glucose control and prevents the onset of hyperglycemia under a high-fat/high-sucrose diet [18]. Conversely, the suppression of IGN promotes the accumulation of lipid in the liver even under a

standard diet. IGN is a recently described function that contributes to the metabolic benefits via the initiation of a gut-brain neural signal, triggering brain-dependent regulation of peripheral metabolism [19]. The authors suggested that IGN can specifically modulate the onset of hepatic steatosis and molecular events associated with the development of NAFLD through a gut-brain-liver neural circuit [18].

References

  1. Overi, D., Carpino, G., Franchitto, A., Onori, P., Gaudio, E. Hepatocyte Injury and Hepatic Stem Cell Niche in the Progression of Non-Alcoholic Steatohepatitis. Cells 2020, 9:pii: E590. doi: 10.3390/cells9030590.
  2. Sun, K., Tordjman, J., Clement, K., Scherer, P. E. Fibrosis and adipose tissue dysfunction. Cell Metab 2013, 18:470-7.
  3. Vily-Petit, J., Soty-Roca, M., Silva, M., Raffin, M., Gautier-Stein, A., Rajas, F., Mithieux, G. Intestinal gluconeogenesis prevents obesity-linked liver steatosis and non-alcoholic fatty liver disease. Gut 2020, Mar 23. pii: gutjnl-2019-319745. doi: 10.1136/gutjnl-2019-319745. 
  4. Soty, M., Gautier-Stein, A., Rajas, F., Mithieux, G. Gut-Brain Glucose Signaling in Energy Homeostasis. Cell Metab 2017, 25:1231-1242.

Reviewer 2 Report

I consider this review a good work, I think it is well written, well organized and describes quite well the role of insulin resistance/T2DM in non alcoholic fatty liver disease progression. References are current. It is a good and current staging of the disease. 

Author Response

Manuscript ID: ijms-792300

Title: The role of insulin resistance and diabetes in nonalcoholic fatty liver disease”.

Point-by-Point Response

We thank the editor and reviewers for the positive assessment of our manuscript and for identifying areas that required correction or modification. Please note that the changes made do not influence the content, conclusions, or framework of the paper. We have not listed below all minor changes made; however, these are indicated in red font in the revised manuscript. All line numbers mentioned in the response to each comment refer to the numbers that appear at the left margin of the text in the revised manuscript.

Reviewer’s Comments to Author:

Reviewer 2

I consider this review a good work, I think it is well written, well organized and describes quite well the role of insulin resistance/T2DM in nonalcoholic fatty liver disease progression. References are current. It is a good and current staging of the disease. 

Thank you for your valuable comment.
